# Detectability and Volumetric Accuracy of Pulmonary Nodules in Low-Dose Photon-Counting Detector Computed Tomography: An Anthropomorphic Phantom Study

**DOI:** 10.3390/diagnostics13223448

**Published:** 2023-11-15

**Authors:** Joost F. Hop, Anna N. H. Walstra, Gert-Jan Pelgrim, Xueqian Xie, Noor A. Panneman, Niels W. Schurink, Sebastian Faby, Marcel van Straten, Geertruida H. de Bock, Rozemarijn Vliegenthart, Marcel J. W. Greuter

**Affiliations:** 1Department of Radiology, University Medical Center Groningen, 9713 GZ Groningen, The Netherlands; anna.walstra@outlook.com (A.N.H.W.); gertjan.pelgrim@mst.nl (G.-J.P.); n.a.panneman@student.rug.nl (N.A.P.); r.vliegenthart@umcg.nl (R.V.); m.j.w.greuter@umcg.nl (M.J.W.G.); 2Department of Radiology, Shanghai General Hospital, Shanghai Jiao Tong University School of Medicine, Shanghai 200080, China; xiexueqian@hotmail.com; 3Siemens Healthineers Nederland B.V., 2595 BN Den Haag, The Netherlands; 4Computed Tomography, Siemens Healthcare GmbH, 91301 Forchheim, Germany; sebastian.faby@siemens-healthineers.com; 5Department of Radiology, Erasmus University Medical Center, 3015 GD Rotterdam, The Netherlands; marcel.vanstraten@erasmusmc.nl; 6Department of Epidemiology, University Medical Center Groningen, 9713 GZ Groningen, The Netherlands; g.h.de.bock@umcg.nl

**Keywords:** computed tomography, X-ray, photon-counting detector, low-dose CT, lung cancer, imaging phantom, pulmonary nodules, dose reduction

## Abstract

The aim of this phantom study was to assess the detectability and volumetric accuracy of pulmonary nodules on photon-counting detector CT (PCD-CT) at different low-dose levels compared to conventional energy-integrating detector CT (EID-CT). In-house fabricated artificial nodules of different shapes (spherical, lobulated, spiculated), sizes (2.5–10 mm and 5–1222 mm^3^), and densities (−330 HU and 100 HU) were randomly inserted into an anthropomorphic thorax phantom. The phantom was scanned with a low-dose chest protocol with PCD-CT and EID-CT, in which the dose with PCD-CT was lowered from 100% to 10% with respect to the EID-CT reference dose. Two blinded observers independently assessed the CT examinations of the nodules. A third observer measured the nodule volumes using commercial software. The influence of the scanner type, dose, observer, physical nodule volume, shape, and density on the detectability and volumetric accuracy was assessed by a multivariable regression analysis. In 120 CT examinations, 642 nodules were present. Observer 1 and 2 detected 367 (57%) and 289 nodules (45%), respectively. With PCD-CT and EID-CT, the nodule detectability was similar. The physical nodule volumes were underestimated by 20% (range 8–52%) with PCD-CT and 24% (range 9–52%) with EID-CT. With PCD-CT, no significant decrease in the detectability and volumetric accuracy was found at dose reductions down to 10% of the reference dose (*p* > 0.05). The detectability and volumetric accuracy were significantly influenced by the observer, nodule volume, and a spiculated nodule shape (*p* < 0.05), but not by dose, CT scanner type, and nodule density (*p* > 0.05). Low-dose PCD-CT demonstrates potential to detect and assess the volumes of pulmonary nodules, even with a radiation dose reduction of up to 90%.

## 1. Introduction

Lung cancer is a major public health concern, accounting for 11.4% of all new cancer cases and an estimated 1.8 million deaths worldwide annually [1]. Because lung cancer is often detected at advanced stages of the disease, it has a low 5-year survival rate of 10–20% [2,3]. Therefore, the role of lung cancer screening using low-dose computed tomography (CT) for the early detection of lung cancer was assessed in several clinical trials [4]. The U.S.-based NLST and Dutch–Belgian NELSON lung cancer screening trials demonstrated that CT screening decreased lung cancer mortality rates by 20% and 24%, respectively [5,6]. As a result, increasing numbers of countries are implementing lung cancer screening programs with low-dose CT [7].

The success of low-dose CT screening is affected by the CT system’s ability to provide sufficient diagnostic image quality for lung nodule detection [8]. An adequate image quality is required to achieve high sensitivity and specificity in detecting nodules of at least 4 mm in diameter (or 30 mm^3^ in volume) and in evaluating nodule growth, which serves as an indicator of malignancy [9,10]. However, radiation also has risks; 10 years of annual CT screening yields an estimated 0.05% additional risk of fatal cancer [11]. To minimize radiation exposure during CT examinations, technical specifications for low-dose CT protocols have been recommended while ensuring adequate image quality [12,13].

The emerging technology of photon-counting detector CT (PCD-CT) may help to address the challenges related to image quality and radiation exposure. PCD-CT has several characteristics that facilitate acquisition with lower noise levels and higher spatial resolution compared to conventional energy-integrating detector CT (EID-CT) [14,15]. Phantom studies are instrumental in assessing CT performance and are particularly valuable because they eliminate the need for human subjects. To assess the CT performance in the imaging of the lung, studies utilize a range of phantoms, from simplified image quality phantoms to more advanced anthropomorphic phantoms [16,17]. In the specific context of evaluating lung nodules, several recent studies have effectively employed an anthropomorphic thorax phantom (Lungman, Kyoto Kagaku, Kyoto, Japan) to evaluate the performance of both EID-CT and PCD-CT systems [18,19]. Recent studies have also shown the potential of PCD-CT to reduce radiation exposure in low-dose lung examinations compared to EID-CT without compromising image quality [20,21]. However, the sensitivity of human observers for the detection of nodules on low-dose PCD-CT remains unknown. Additionally, the volumetric accuracy of clinically relevant nodules of different shapes has not been established for PCD-CT.

The aim of this study was therefore to assess the sensitivity of the detection of pulmonary nodules and volumetric accuracy with PCD-CT at low-dose levels compared to EID-CT. To achieve this, lung cancer screening participants were simulated using an anthropomorphic phantom.

## 2. Materials and Methods

### 2.1. Phantom and Nodules

An anthropomorphic thorax phantom was used, extended with a tissue-equivalent extension layer in order to resemble an average-weighted Western male screenee (Lungman). The phantom consisted of a thoracic cavity in which an insert with artificial vascular structures and mediastinum was placed.

In total, 35 artificial pulmonary nodules were developed in-house: 11 spherical, 12 lobulated, and 12 spiculated nodules. Each shape was made in 2 different densities to resemble subsolid and solid nodules: −330 ± 50 Hounsfield Units (HU) and 100 ± 30 HU, respectively, determined with EID-CT at 100 kVp with 0.4 mm tin (Sn) filter. The nodules were manufactured in different sizes, with diameters ranging from 2.5 to 10 mm and corresponding volumes ranging from 5 to 1222 mm^3^. Additionally, one commercially available synthetic nodule of 10 mm was used (Kyoto Kagaku, Kyoto, Japan). An overview of the phantom setup is presented in Figure 1.

Three-dimensional models of spherical, lobulated, and spiculated nodules were obtained from a previous study [22]. The subsolid nodules were created through casting of polyurethane foam in 3D-printed molds, while the solid nodules were directly 3D-printed (Form 3B+, Formlabs, MA, USA). To enable easy localization after phantom insertion, the subsolid nodules were colored red or blue, while the solid nodules were UV-reflective.

### 2.2. CT Systems and Protocols

A clinical PCD-CT system (NAEOTOM Alpha, Siemens Healthineers, Erlangen, Germany) and a state-of-the-art EID-CT system (SOMATOM Force, Siemens Healthineers) were used. With EID-CT, a reference low-dose lung cancer screening protocol was applied: 100 kVp with Sn-prefiltration (Table 1) [12,13]. With PCD-CT, acquisition and reconstruction parameters were selected to resemble the reference EID-CT protocol. Both scanners employed automatic tube current modulation, referred to as quality reference milliampere-seconds (QRM) and image quality (IQ) level with EID-CT and PCD-CT, respectively.

However, there were notable differences between EID-CT and PCD-CT protocols. PCD-CT images were synthesized as virtual monoenergetic images (VMI) at 70 keV to align with the mean X-ray energy of photons at Sn100 kVp in EID-CT [23]. For PCD-CT, a regular body kernel (Br40) for nodule detection and an additional sharp quantitative kernel (Qr60) for volume assessment were used to match corresponding EID-CT kernels, Br40 and Qr59. Moreover, PCD-CT featured an automatic adjustment of the pixel matrix size. This adjustment depended on the selected field-of-view (FOV) and reconstruction kernel. Matrix sizes of 512 × 512 and 768 × 768 were automatically selected for images reconstructed with a Br40 and Qr60 kernel, respectively [24]. With PCD-CT, the images were reconstructed using quantum iterative reconstruction at strength level 3 (QIR-3) compared to iterative reconstruction at strength level 3 (ADMIRE-3) with EID-CT. It has been demonstrated that QIR-3 resulted in high overall image quality for low-dose PCD-CT chest scans [25].

### 2.3. CT Acquisitions

The phantom was scanned once on the EID-CT system to establish a reference dose value. The resulting CTDI_vol_ was recorded. Subsequently, the PCD-CT scans were conducted at five dose levels of 100%, 75%, 50%, 25%, and 10% of the reference dose, achieved by setting the IQ level to 12, 9, 6, 3, and 1, respectively.

The nodules were manually placed in the lung insert of the phantom, alternating between the left and right lung; the location of each nodule was registered. All nodules were attached to pulmonary vessel structures, but not to pleural or mediastinal structures. All 36 nodules were scanned 3 times as per CT protocol with varying positions within the phantom. To achieve this, a randomization scheme was generated using an in-house developed software tool, with 20 setups with 0 to 12 nodules. This resulted in a total of 120 acquisitions, and as a result of using two reconstruction kernels, 240 image series were reconstructed. To mimic inter-scan displacement, the phantom underwent a sub-centimeter translocation and rotation between each nodule setup.

To serve as ground truth, the physical volumes of the synthetic nodules were derived by segmentation of high-resolution images acquired on a preclinical micro-CT scanner (Inveon, Siemens Healthineers). The nodules were scanned in air at a tube voltage of 40 kVp and a current of 250 µA. Scan duration was 30 min and 24 s. The micro-CT images had a voxel size of 59 µm. From the micro-CT images, the nodules were segmented in 3D using a grow-from-seeds algorithm, and then, the volumes of the 3D segmentations were calculated (3D analysis and visualization software Slicer, version 5.2.1) [26].

### 2.4. Image Analysis

To assess nodule detectability, two experienced observers (observer 1—GP, with more than 10 k low-dose chest CT cases reviewed for lung nodules during 4 years, and observer 2—XX, with over 20 years of experience in thoracic radiology) were blinded to the nodules’ presence and location. They independently assessed all images in axial and coronal plane, displayed as 1.0 mm thin slices, and maximum intensity projection (MIP) with 10 ± 0.4 mm slice thickness (MM Reading, syngo.via, version 2.0.8, Siemens Healthineers).

To assess volumetric accuracy, a third trained observer (observer 3–NP) measured the volumes of all nodules detected by the first and/or second observer. As each nodule was scanned 3 times per CT protocol and not all nodules were detected by the observers, the number of measurements per nodule ranged from 0 to 3. Measurements were performed in axial images using the ‘Solid Lesion’ quantification mode of a semi-automatic software tool (Lesion Quantification, syngo.via, version 2.0.8, Siemens Healthineers). When appropriate, manual adjustment of the automatically generated segmentations was performed using the brush tool.

### 2.5. Statistical Analysis

Detectability of nodules was calculated as the sensitivity per observer for each nodule with PCD-CT at different dose levels and with EID-CT. Mean sensitivity of PCD-CT and EID-CT for spherical, lobulated, and spiculated nodules were calculated. The inter-observer agreement between observer 1 and 2 was determined using Cohen’s Kappa. Kappa values larger than 0.80 were considered as strong agreement, 0.61–0.80 as substantial agreement, and ≤0.60 as weak [27].

Volumetric accuracy was defined as the percentage error between physical and observed volumes. The percentage error was calculated for all detected nodules using the following equation:$$Percentage\;error=100\%\times \frac{V_{observed}-V_{physical}}{V_{physical}}$$
where V_observed_ and V_physical_ represent the observed and physical nodule volumes, respectively. Median absolute percentage errors were calculated and compared for PCD-CT and EID-CT using a two sample *t*-test. Intra-observer variability in volume measurements was assessed using the intraclass correlation coefficient (ICC) for nodules that were measured more than once.

To assess the factors influencing nodule detectability and volumetric accuracy, multivariable analyses were performed using a general linear regression model. Sensitivity and volumetric accuracy served as dependent variables, while the independent variables included CT scanner type (EID-CT or PCD-CT), PCD-CT dose level, observer (observer 1 and 2), physical nodule volume, nodule shape, and nodule density. Observer impact on volumetric accuracy was not evaluated, as a single observer performed all measurements.

The outcome of the statistical test (*t*-test) in the multivariable analysis included the *t*-value and *p*-value. The higher the *t*-value, the greater the likelihood that a factor influences the dependent variable. When applicable, the highest *t*-values were reported for independent variables. Statistical significance was considered for *p* < 0.05. Statistical analyses were performed using R Statistical Software (Foundation for Statistical Computing, v.4.1.0, Vienna, Austria).

## 3. Results

### 3.1. Detectability

Figure 2 displays the representative thorax phantom images from EID-CT and PCD-CT. In 120 CT examinations, 642 nodules were present. This count included 36 nodules that were scanned three times across six CT protocols (five PCD-CT dose levels and one EID-CT protocol), as well as one nodule that was unintentionally scanned twice instead of three times. Observer 1 detected 367 nodules (57%), while observer 2 found 289 nodules (45%), with 383 nodules (60%) detected overall. The false-positive rates were 6.9% for observer 1 and 1.4% for observer 2. The inter-observer agreement was substantial (κ = 0.65).

The nodule detectability varied by size and shape. At a 100% dose, PCD-CT demonstrated comparable performance to EID-CT in terms of nodule detection (Figure 3). With PCD-CT, observer 1 detected all spherical, lobulated, and spiculated nodules with volumes of at least 70 mm^3^, 77 mm^3^, and 115 mm^3^, respectively, and with EID-CT with volumes of at least 70 mm^3^, 37 mm^3^, and 818 mm^3^. Observer 2 detected all nodules with PCD-CT larger than 118 mm^3^, 77 mm^3^, and 727 mm^3^, respectively, and larger than 138 mm^3^, 122 mm^3^, and 727 mm^3^ with EID-CT. Observer 1 had a mean sensitivity for all spherical, lobulated, and spiculated nodules of 56%, 53%, and 69%, respectively; with EID-CT, the sensitivity was 61%, 56%, and 58%, respectively. Observer 2 showed mean sensitivities for spherical, lobulated, and spiculated nodules of all sizes of 56%, 39%, and 47% with PCD-CT and, respectively, 47%, 47%, and 50% with EID-CT.

The sensitivity of nodule detection with PCD-CT was unaffected by the reduction in the dose settings, even down to 10% of the reference dose (*p* > 0.05) (Figure 4). Also, the CT scanner type and nodule density had no significant impact (*p* > 0.05). However, the detectability of nodules was significantly influenced by the observer, nodule volume, and a spiculated nodule shape (*p* < 0.05). An overview of the multivariable analysis findings is provided in Table 2.

### 3.2. Volumetric Accuracy

Figure 5 presents the digital model used for the 3D printing of the lobulated nodules and an example micro-CT image with the corresponding segmentation from which the physical volume was calculated. All 383 detected nodules were successfully segmented: 135 spherical nodules (15–954 mm^3^), 110 lobulated nodules (17–389 mm^3^), and 138 spiculated nodules (18–1222 mm^3^). The intra-observer agreement for volumetric measurements was excellent for all nodule volumes, shapes, and densities, with ICCs above 0.93.

Both PCD-CT and EID-CT showed an underestimation of the nodule volumes, except for one small spiculated nodule of 18 mm^3^, which was overestimated (Figure 6). The median absolute percentage errors among all nodules were 20% with PCD-CT and 24% with EID-CT at an equal dose (*p* > 0.05). For spherical, lobulated, and spiculated nodules, specifically, the median absolute percentage errors were 15%, 18%, and 25%, respectively, with PCD-CT, and 24%, 22%, and 28% with EID-CT. With PCD-CT, the volumes of the spherical nodules were underestimated by 10–33%, lobulated nodules by 8–41%, and spiculated nodules by 20–52%. With EID-CT, the spherical nodules were underestimated by 9–33%, lobulated nodules by 11–41%, and spiculated nodules by 14–52%.

As for detectability, the volumetric accuracy was unaffected by a radiation dose reduction with PCD-CT down to 10% (*p* > 0.05) (Table 2). The scanner type and nodule density had no significant influence on the volumetric accuracy (*p* > 0.05). The physical nodule volume and spiculated shape were identified as significant factors influencing the volumetric accuracy (*p* < 0.05).

## 4. Discussion

In this phantom study, the detectability and volumetric accuracy of artificial pulmonary nodules were evaluated with PCD-CT at varying low-dose levels and compared to conventional EID-CT. This is the first study to assess the sensitivity of human observers for the detection of nodules with PCD-CT and to assess the volumetric accuracy for clinically relevant nodules larger than 30 mm^3^.

An important finding is that neither the detectability nor volumetric accuracy were significantly affected by reducing the PCD-CT radiation dose by up to 90% with respect to EID-CT, corresponding to a CTDI_vol_ of 0.07 mGy. Furthermore, the use of either PCD-CT or EID-CT, as well as variations in the nodule density, did not have a significant effect on the detectability or volumetric accuracy either. However, a spiculated nodule shape and the nodule size significantly influenced both the detectability and volumetric accuracy. Moreover, for closely matched low-dose PCD-CT and EID-CT protocols, PCD-CT demonstrated similar detectability. Both PCD-CT and EID-CT consistently underestimated the nodule volumes, with PCD-CT showing a lower median absolute percentage error across all the nodule shapes.

This study’s findings align with prior research conducted on EID-CT, which investigated the impact of dose reduction through the spectral shaping of the X-ray beam with a tin filter to sub-milligray levels. These studies reported no significant decrease in the nodule detectability and volumetric accuracy, consistent with our observations with PCD-CT [28,29,30]. In a previous phantom study focusing on PCD-CT, a comparable nodule detectability and volumetric accuracy to EID-CT were demonstrated using an AI-based computer-aided diagnosis (CAD) system [19]. However, this study only included radiation dose levels down to approximately 50% of the reference dose used in this study. Additionally, a direct comparison with human readers was not conducted. Another phantom study assessed the lung nodule detectability on an experimental PCD-CT system from a different vendor [16]. This study employed a task-based model observer, a mathematical model used to predict human reader performance in lesion detection, primarily based on image noise and spatial resolution. Their results indicated a notably higher detectability with PCD-CT compared to EID-CT at relatively high dose levels (CTDI_vol_) of 3.9 mGy. However, such discrepancies in nodule detectability were not observed in this study; rather, we found similar levels of detectability between the two scanner types. This can be attributed to the fact that EID-CT and PCD-CT were used under closely matched low-dose protocols, maintaining consistent in-plane resolution. Also, images of diagnostic quality, as used in their study, possess different noise properties compared to low-dose images. Another phantom study reported similar results in terms of the volumetric accuracy, with some differences attributed to the phantom design and scan protocols [31]. They observed comparable volumetry between EID-CT and PCD-CT at similar low dose levels, and a consistent underestimation of the nodule volumes, aligning with our findings. Their study’s limitation was the use of an idealized homogeneous phantom background for nodule measurements, while in this study, the measurements were conducted in a heterogeneous background. Moreover, their study only included nodules larger than 38 mm^3^, whereas, in our study, nodules as small as 15 mm^3^ were measured. In our study, we found substantial inter-observer agreement (κ = 0.65) regarding nodule detection, while recognizing that the individual observer significantly influences nodule detectability. This finding aligns with findings from a sub-study of the NLST lung cancer screening trial, where substantial inter-observer agreement (κ = 0.64) was found among 16 experienced observers for detecting nodules with a size threshold of 4 mm (or 30 mm^3^) [32]. However, considerable variation among individual pairs of readers was demonstrated (κ = 0.40–0.82). This variation arose not only due to differences in nodule detection and interpretation (distinguishing nodules from non-nodules) but also from variations in how different observers determined nodule sizes at the 4 mm threshold [33]. Promisingly, the integration of CAD has shown potential to reduce the inter-observer variability and improve detection substantially [34].

The findings from this study imply that PCD-CT has the potential to be implemented in a lung cancer screening setting, even at very low-dose levels compared to the currently recommended protocol guidelines [12,13]. PCD-CT has previously been reported to offer superior image quality to EID-CT for lung cancer screening examinations, despite lower dose [35]. The improvement in image quality with PCD-CT can be attributed to differences in detector technology and reconstruction techniques. In conventional EID-CT systems, incident X-ray photons are converted to visible light in the scintillation layer of the detector and then transformed into an electrical signal. In contrast, in PCD-CT systems, incident photons are directly converted into an electrical signal, resulting in the elimination of electronic noise. Furthermore, the detector elements in PCD-CT are manufactured smaller in size thanks to the absence of separate scintillator elements and isolating septa, resulting in a higher intrinsic spatial resolution compared to EID-CT [15]. However, despite these theoretical improvements, the outcomes from this study imply that the inherent differences between PCD-CT and EID-CT, without protocol adjustments, do not significantly affect the detectability and volumetric accuracy.

The current nodule management guidelines for lung cancer screening define criteria for detecting and evaluating (sub)solid nodules [9,10]. These guidelines define minimum nodule volume requirements to be detected at a baseline of at least 100 mm^3^, and for incidental nodules during follow-up, a minimum of 30 mm^3^. Screenees with such nodules become eligible for increased follow-up CT scans. The accurate assessment of nodule volume is crucial for evaluating their growth, often quantified as volume doubling time. Published claims specify the required accuracy when measuring nodule volumes, suggesting that solid nodules ranging from 113 to 905 mm^3^ should be measured, with values falling within a 95% CI range of ±59% to ±22% accuracy [36]. Considering our study’s findings for nodule detectability, our results largely meet the sensitivity requirements outlined in the current nodule management guidelines for detecting clinically relevant nodules at baseline. However, it is important to note that our study showed variable sensitivity for detecting small nodules ranging between 30 and 100 mm^3^, regardless of whether EID-CT or PCD-CT was used, with sensitivities ranging from 0% to 100%. Regarding volumetric accuracy, our study identified that volumes were consistently underestimated, with PCD-CT showing underestimations of 15% to 25%, which aligns with the required accuracy. Notably, EID-CT exhibited more substantial underestimation, ranging from 22% to 28%. Although this difference in volume underestimation was not statistically significant, it suggests potential improvements in volume measurements with PCD-CT compared to EID-CT.

This study has some limitations. First, the phantom used in this study lacked parenchyma and airways, leading to a representation that is clinically less realistic. Second, the observers’ prior training with EID-CT images could introduce bias when assessing PCD-CT scans. Third, we did not perform a dose reduction study with EID-CT to assess the influence on detectability and volumetric accuracy. However, other studies have assessed the influence of radiation dose reduction on nodule detection and volumetric accuracy with EID-CT, indicating no significant decrease in nodule detectability and volumetric accuracy [30,31,37]. Lastly, the study’s assessment of spiculated nodules is incomplete, due to a significant data gap within the volume range of nodules between 115 and 727 mm^3^. Also, the artificial spiculated nodules, created through a printing process, may not fully replicate the characteristics of real-life spiculated nodules, as they exhibited a more coarsely spiculated morphology.

In this study, the performance of low-dose PCD-CT in terms of the detectability and volumetric accuracy of small pulmonary nodules with varying shapes and densities was assessed. Further research involving human subjects and protocol optimization is necessary to validate and improve the usability of PCD-CT in lung cancer screening programs. While low-dose CT screening is an essential tool for the early detection of lung cancer, its effectiveness in characterizing suspicious nodules is constrained by its reliance solely on image-derived nodule features, lacking valuable information on pathological biomarkers. It is therefore important to explore alternative techniques that can improve the early detection of malignant lung nodules. One promising approach involves minimally invasive methods, such as liquid biopsy, which can complement the characterization of suspicious nodules during follow-up in screening programs, providing additional information beyond CT-derived characteristics [38,39].

In conclusion, low-dose PCD-CT demonstrated great potential for the detection and volumetric assessment of pulmonary nodules. It performed similarly to EID-CT, even with a 90% reduction in the radiation dose compared to the current recommended lung cancer screening protocols.

## Figures and Tables

**Figure 1 diagnostics-13-03448-f001:**
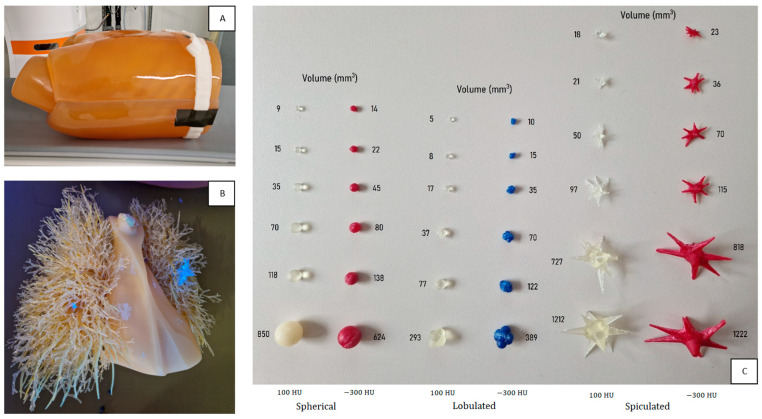
Overview of the phantom setup. (**A**) Anthropomorphic thorax phantom with tissue-equivalent chest plates, (**B**) 2 UV-reflective nodules inserted in the lungs, and (**C**) an overview of the 36 artificial nodules of different sizes, shapes, and densities. CT values of the subsolid (red and blue) and solid (translucent white) nodules were −330 ± 50 HU and 100 ± 30 HU, respectively. Volumes ranged from 5 to 1222 mm^3^.

**Figure 2 diagnostics-13-03448-f002:**
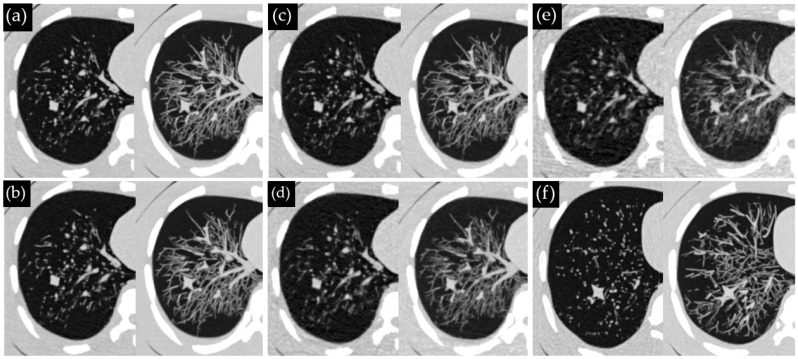
Visual representation of CT image slices of the thorax phantom acquired using PCD-CT at (**a**) 100%, (**b**) 75%, (**c**) 50%, (**d**) 25%, (**e**) 10% of the reference dose and (**f**) EID-CT. The left and right images in each sub-figure represent 1.0 mm axial slices and 10 mm maximum intensity projection slices, respectively.

**Figure 3 diagnostics-13-03448-f003:**
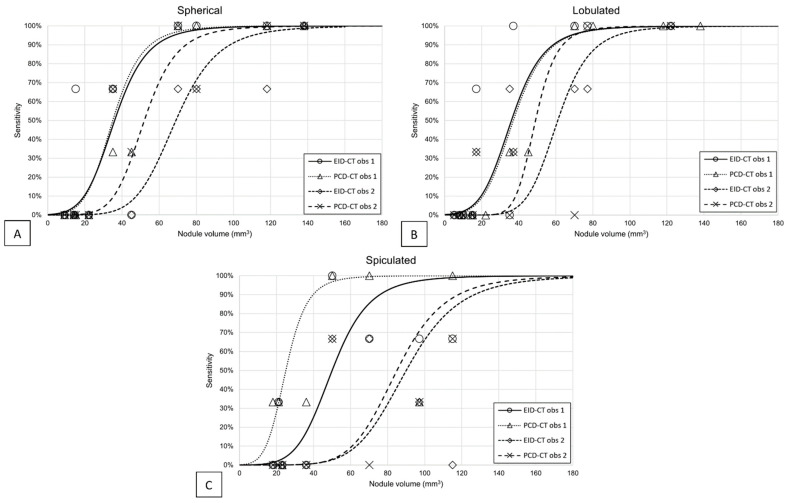
Sensitivity for detection of (**A**) spherical, (**B**) lobulated, and (**C**) spiculated nodules as a function of nodule volume with PCD-CT at 100% dose and EID-CT. The lines represent sigmoid curves through the sensitivity data of observer 1 and 2 (obs 1 and obs 2). Sensitivity data of solid and subsolid nodules were aggregated.

**Figure 4 diagnostics-13-03448-f004:**
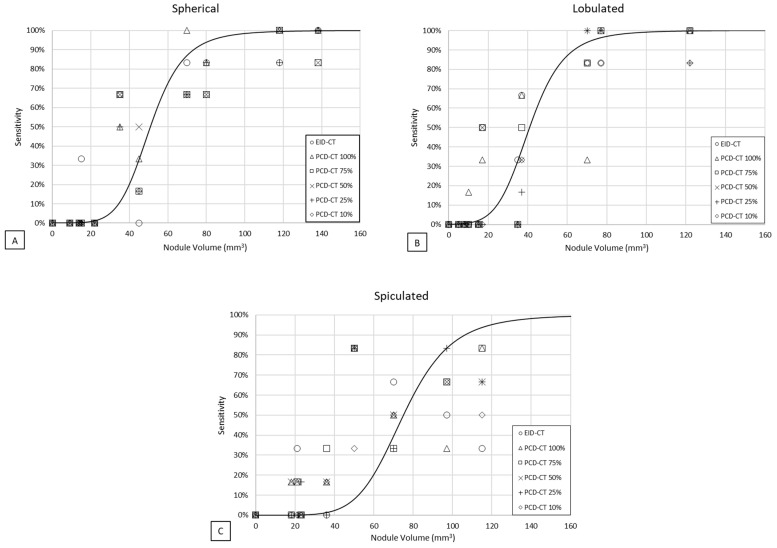
Mean detection sensitivity as a function of nodule volume for the detection of (**A**) spherical, (**B**) lobulated, and (**C**) spiculated nodules with EID-CT and PCD-CT at 100%, 75%, 50%, 25%, and 10% PCD-CT dose with respect to the reference EID-CT dose. Data points were averaged over observer and sigmoid curves were fit through all points.

**Figure 5 diagnostics-13-03448-f005:**
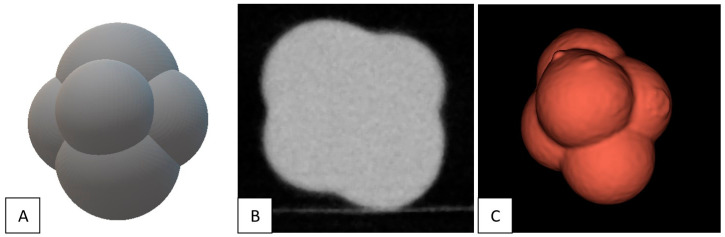
Visual representation of (**A**) the digital model used for the 3D printing of lobulated nodules, (**B**) an example micro-CT image of a 293 mm^3^ solid nodule, and (**C**) the corresponding nodule segmentation derived from the micro-CT image, used for calculation of physical volume.

**Figure 6 diagnostics-13-03448-f006:**
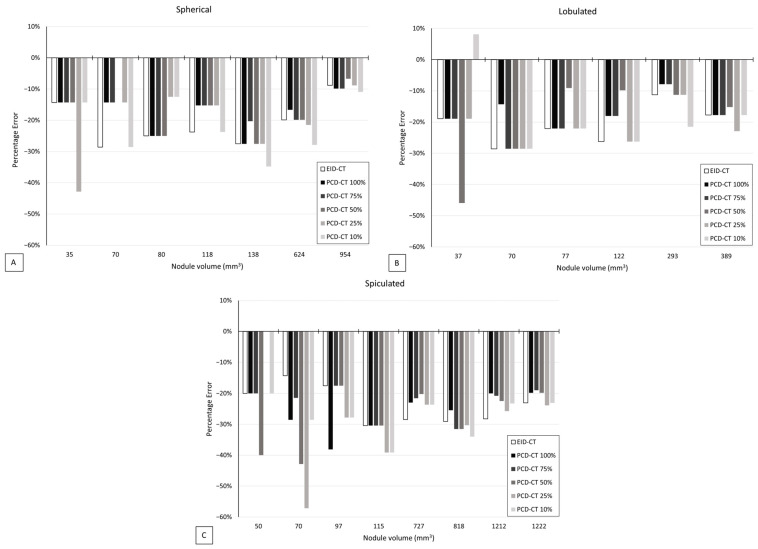
Volumetric accuracy presented as percentage error between physical and observed volumes with EID-CT and with PCD-CT at 10–100% dose for (**A**) spherical, (**B**) lobulated, and (**C**) spiculated nodules. Only nodules that were measured on both EID-CT and PCD-CT were incorporated in this figure.

**Table 1 diagnostics-13-03448-t001:** Comparison of the energy-integrating CT (EID-CT) and photon-counting CT (PCD-CT) protocols. The reference EID-CT protocol used a low-dose lung cancer screening approach. The low-dose PCD-CT protocol aimed to match the scan parameters of the reference protocol while allowing for variations in dose levels ranging from 10% to 100% of the reference dose, achieved through adjustments to the image quality (IQ) level.

Protocol	EID-CT	PCD-CT
Scan mode	Spiral	Spiral
Tube voltage (kVp)	Sn100 (0.6 mm Sn)	Sn100 (0.4 mm Sn)
QRM/IQ level	187 (QRM)	1, 3, 6, 9, 12 (IQ)
CTDI_vol_ (mGy)	0.79	0.07, 0.20, 0.41, 0.61, 0.81
Rotation time (s)	0.5	0.5
Pitch	1.2	1.2
Detector configuration	192 × 0.6 mm	144 × 0.4 mm
Slice thickness/increment (mm)	1.0/0.7	1.0/0.7
FOV (mm)	300	300
VMI reconstruction	-	70 keV
Reconstruction kernel	Br40, Qr59	Br40, Qr60
Image matrix size	512 × 512	512 × 512, 768 × 768
Reconstruction algorithm and strength	ADMIRE-3	QIR-3

**Table 2 diagnostics-13-03448-t002:** Factors influencing nodule detectability and volumetric accuracy assessed by multivariable analysis.

	Dependent Variables			
Independent Variables	Sensitivity		Volumetric Accuracy	
	*t*-Value	*p*	*t*-Value	*p*
Scanner type	0.12	0.91	1.04	0.31
PCD-CT dose	−0.93	0.35	1.52	0.13
Observer	−3.49	<0.001	N/A	N/A
Physical volume	14.82	<0.001	2.03	<0.05
Shape (spiculated)	−2.51	<0.05	−2.18	<0.05
Density	1.00	0.32	1.52	0.13

## Data Availability

Data are contained within the article.

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
