# Peer review of "Detectability and Volumetric Accuracy of Pulmonary Nodules in Low-Dose Photon-Counting Detector Computed Tomography: An Anthropomorphic Phantom Study"

_diagnostics, 2023, doi:10.3390/diagnostics13223448_

Round 1

Reviewer 1 Report

Comments and Suggestions for Authors

This is an excellent an excellent study, which is of high importance for the specialists involved in CT-based lung cancer diagnostics and screening. The authors may consider some additional issues in order to attract wider audience of medical professionals:

Brief introductory overview on phantom studies may be inserted in the Introduction.

Relevant studies addressing interobserver variability may be mentioned in the Discussion.

Limitation of imaging technologies deserve to be acknowledged in the Discussion; for example, there are promising studies which utilized “liquid biopsy” (plasma ctDNA or microRNA testing) for subjects with suspicious nodules.   

Reviewer 2 Report

Comments and Suggestions for Authors

This reviewer thinks that this article is one of the best written articles that he has ever reviewed..! Congratulations to the authors.

Great description on the methods used and great presentation of the results.!
